# The Cactus (*Opuntia ficus-indica*) Cladodes and Callus Extracts: A Study Combined with LC-MS Metabolic Profiling, In-Silico, and In-Vitro Analyses

**DOI:** 10.3390/antiox12071329

**Published:** 2023-06-23

**Authors:** Dong-Geon Nam, Hee-Sun Yang, Ui-Jin Bae, Eunmi Park, Ae-Jin Choi, Jeong-Sook Choe

**Affiliations:** 1Division of Functional Food & Nutrition, Department of Agrofood Resources, National Institute of Agricultural Science, Rural Development Administration, Wanju-gun 55365, Republic of Korea or realfoods@korea.kr (D.-G.N.); hs0704@korea.kr (H.-S.Y.); euijin0230@korea.kr (U.-J.B.); aejini77@korea.kr (A.-J.C.); 2Department of Food and Nutrition, Hannam University, Daejeon 306-791, Republic of Korea; eunmi_park@hnu.kr

**Keywords:** *Opuntia ficus-indica* cladodes, callus, ethanolic extract, phytochemical, LC-MS, molecular docking, antioxidant, anti-inflammatory

## Abstract

*Opuntia ficus-indica* (OF) phytochemicals have received considerable attention because of their health benefits. However, the structure-activity relationship between saponin and flavonoid antioxidant compounds among secondary metabolites has rarely been reported. In a molecular docking study, selected compounds from both *Opuntia ficus-indica* callus (OFC) and OF ethanol extract were found to be involved in Toll-like receptor 4 and mitogen-activated protein kinase (MAPK) signaling pathways. High affinity was specific for MAPK, and it was proposed to inhibit the oxidative and inflammatory responses with poricoic acid H (−8.3 Kcal/mol) and rutin (−9.0 Kcal/mol). The pro-inflammatory cytokine factors at a concentration of 200 μg/mL were LPS-stimulated TNF-α (OFC 72.33 ng/mL, OF 66.78 ng/mL) and IL-1β (OFC 49.10 pg/mL, OF 34.45 pg/mL), both of which significantly decreased OF (*p* < 0.01, *p* < 0.001). Taken together, increased NO, PGE_2_, and pro-inflammatory cytokines were significantly decreased in a dose-dependent manner in cells pretreated with OFC and the OF extract (*p* < 0.05). These findings suggest that OFC and OF have important potential as natural antioxidant, anti-inflammatory agents in health-promoting foods and medicine.

## 1. Introduction

Today’s population is approximately 7.96 billion people, and about 50% of the population is concentrated in warmer regions (20–60° north latitude). Modern human ecology is polarized according to the geography of the earth due to radical industrialization since the 1960s [1]. This unbalanced population distribution is facing a singularity in the ecological environment. In addition, environmental changes caused by global warming will affect land agriculture in the future by causing phenomena such as land flooding due to sea level rises and increased frequency and intensity of heat waves due to surface temperature rises. Environmentally friendly soilless cultivation technology is a promising approach for these problems of future agriculture [2].

Plant cell culture is a cultivation technology that can aseptically culture plant cells and organs using a medium containing nutrients, and it can uniformly mass-produce throughout the year without soil [3]. Populations of single cells, regenerating into fully functional plants, are grown in containers filled with a variety of nutrient-rich growing media [4]. Cultivation methods can improve environmental control and avoid uncertainties in soil water and nutrient status. Callus cultures are plant tissues of improved quality that are formed as meristems masses derived from plant wounds or plant cells. These transgenic plants can be used to confirm the function of a specific gene or to cultivate new crop varieties with agriculturally useful traits, and callus cultures extracts have been in the spotlight as useful materials that can be used for cosmetics or food.

Two species of Opuntia are used as food in Korea (*O. ficus-indica* and *O. humifusa*). *O. ficus-indica* is a naturalized cactus on Jeju Island, Republic of Korea (originating from Mexico), and its fruit is called nopal (*baeknyeoncho* in Korean). *O. humifusa* grows in Gyeonggi-do, Republic of Korea. The Korean names are *opuntieae, baeknyeoncho, bogum,* and *palm*, which were coined in various forms influenced by lifestyles. These cacti are used as food after processing them into dried powder, extract, or concentrate or after fermentation, and the fruit is used for oil and beverages. Cacti are plants specialized for a dry environment. They adapt according to the influence of extreme natural environmental factors (light, seasonal change, temperature, moisture, disease, insects, soil, etc.). In addition, substances useful to humans (phytochemicals) have been obtained. Despite being a potential nutraceutical, the parts used (cladode and fruit) are limited.

As an important reservoir of various biologically active compounds, called phytochemicals, plants are essential for treating several human diseases [5]. Phytochemicals are chemical products, including defense factors, that are metabolized by all plants when they are exposed to various environmental factors (ultraviolet rays, pests, microorganisms, etc.) from seed to adult growth [6]. Growing plants can synthesize and accumulate a comprehensive spectrum of chemical compounds in response to physiological stimuli and stress [7]. Representative components of phytochemicals include saponin, phenolic acid, flavonoids, and carotenoids. Plants with these functional components are used for food and medicine and have been highlighted as useful raw materials for healing various defects in the human body.

In this study, a molecular docking study was performed to determine whether cactus and callus extracts could competitively bind to Toll-like receptor 4 and mitogen-activated protein kinase and block lipopolysaccharide-induced inflammation. Anti-inflammatory activity and phytochemical components of cactus and callus were identified and observed.

## 2. Materials and Methods

### 2.1. Plant Materials and Reagents

Plant materials: *Opuntia ficus-indica* (OF) plant material, i.e., mature cladodes as a dried powder, was purchased from Jeju Island of Republic of Korea (33°22′30.5″ N 126°12′57.0″ E, m.jeju100.com (accessed on 19 May 2020), 5 years old). To induce *Opuntia ficus-indica* callus (OFC) growth, OF was first disinfected in a 5 mM salicylic acid solution for 2 min, 70% ethanol for 10 min, and 1% bleach (sodium hypochlorite) for 10 min and then was washed three times in sterile distilled water for 15 min. Sterilized OF was cut into 1 × 1 cm^2^ pieces and inoculated into media. For media preparation, agar (8 g/L), sucrose (10.09 g/L), and 3-(N-morpholino) propane sulfonic acid (MOPS; MB Cell Company, Seoul, Republic of Korea) (0.5 g/L) were added to 2,4-dichlorophenoxy acetic acid medium (2,4-D medium; MB cell Company, Seoul, Republic of Korea) (1 mg/L). The media were adjusted to a pH of 5.7 and autoclaved at 121 °C for 15 min before use. Sterilized OF was cultivated in media at room temperature for about 4 weeks. Calluses were then harvested and cultured with liquid medium without agar at room temperature for 2 weeks. OFC extracts were then filtered, vacuum-concentrated, and freeze-dried in the experiments described below. Samples were stored at −70 °C for further experimental work.

Reagents: Dulbecco’s modified Eagle’s medium (DMEM), fetal bovine serum (FBS), penicillin-streptomycin (pen-strep), and phosphate-buffered saline (PBS) were purchased from Grand Island Biological Company (Grand Island, NY, USA). Lipopolysaccharide (LPS) and dimethyl sulfoxide (DMSO) were purchased from Sigma-Aldrich (St. Louis, MO, USA), the EZ-Cytox cell viability assay kit from DaeilBio (Seoul, Republic of Korea), and the Griess reagent from Promega Co. (Madison, WI, USA). The enzyme-linked immunosorbent assay (ELISA) kits for cytokines and PGE_2_ were purchased from R&D Systems Inc. (Minneapolis, MN, USA).

### 2.2. Sample Preparation

The weighed OF and OFC dried powder (1.0 g) were suspended in 10 mL of 70% ethanol containing the extract on a bullet blender tissue homogenizer (Next Advance, NY, USA). After centrifugation (3000 rpm, 4 °C, 15 min), the supernatant was immediately filtered using a syringe filter (PVDF 0.20 μm, Whatman, Kent, United Kingdom), and then 0.5 mL of the filtrate was diluted with water to a final volume of 5 mL. The experiments were performed in triplicate.

### 2.3. UPLC-QTOF-MS Metabolic Profiling

These samples were analyzed using Waters Acquity UPLC-QTOF. The sample extract was injected into an Acquity UPLC BEH C18 column (2.1 mm × 100 mm, 1.7 μm; Waters), and the mobile phase was: (A) water containing 0.1% formic acid; or (B) acetonitrile containing 0.1% formic acid. The flow rate was 0.35 mL/min, the analysis time was 12 min, and the column temperature was 40 °C. The gradient elution had the following profile: 0–0.01 min, 99% B; 0.01–1.0 min, 99% B; 1.0–8.0 min, 100% B; 8.0–9.0 min, 100% B; 9.0–9.5 min, 99% B; 9.5–12.0 min, 99% B. The eluents passing through the column were analyzed by QTOF-MS with a negative or positive electrospray ionization (ESI) mode. The QTOF-MS conditions in negative and positive modes were capillary and sampling cone voltages of 2.5 kV/3 kV and 20 V/40 V, respectively, a desolvation flow rate of 900 L/h/800 L/h, a desolvation temperature of 400 °C, and source temperature of 100 °C. QTOF-MS data were analyzed at a scan range of 100–1500 *m*/*z* and a scan time of 0.2 s, and leucine-enkephalin was used as a reference compound for the lock mass. MS/MS spectra were obtained under the conditions of a collision energy ramp (10–30 eV) with *m*/*z* of 50–1500. Mass spectrometry data processing, including *m*/*z*, retention time, and ion intensity, was performed using UNIFI software, version 1.8.2.169 (Waters). Material identification was conducted through online databases connected to UNIFI software.

### 2.4. In-Silico Prediction of Molecular Docking Simulation

#### 2.4.1. Ligand and Target Protein Preparation

In this study, only eight of the bioactive compounds identified in OFC and OF extracts by LC-MS analysis were selected for docking studies. Chemical structures of selected bioactive compounds were retrieved through NCBI’s PubChem compound database. Two-dimensional structures were obtained from NCBI’s PubChem compound database, and ligands were prepared using UCSF Chimera. Before proceeding with docking assays, Toll-like receptor 4 (TLR4, PDB ID: 2Z62) and mitogen-activated protein kinase (MAPK, PDB ID: 5MTY) proteins were purified and energy optimized. The study was converted into a protein model using UCSF Chimera software (version 1.16). Finally, PyMol software (version 2.1.0) was used to analyze the protein structures, hydrogen-bonding interactions, and non-bonding interactions of ligands and active site residues and to prepare high-resolution images.

#### 2.4.2. Molecular Docking

UCSF Chimera and AutoDock-Vina software were used to dock the anti-inflammatory target proteins TLR4 and MAPK with selected functional compounds and calculate binding energies. Hydrogen atoms, force fields (CHAEMm), and atomic charges were added. The binding energy and binding contacts of each ligand were obtained, and the docked complex was analyzed using PyMol.

### 2.5. In-Vitro Anti-Inflammatory Activity

#### 2.5.1. Cell Culture and Viability Assay

RAW 264.7 murine macrophage cells (Korean Cell Line Bank, Seoul, Republic of Korea) were cultured using DMEM containing 10% FBS and 1% pen–strep solution and maintained in a humidified incubator at 37 °C with 5% carbon dioxide (CO_2_). The cells were seeded into a 96-well plate (4 × 10^4^ cells/well) and incubated for 24 h. Samples were dissolved in DMSO and incubated with the cells at various doses for an additional 24 h. Subsequently, 0.1 µL/mL of EZ-Cytox solution was added to each well and incubated at 37 °C for 3 h. Then, the absorbance was confirmed at 450 nm on a microplate reader (Infinite M200 Pro; Tecan, Krems, Austria).

#### 2.5.2. Nitric Oxide (NO) Assay

The cells were seeded into a 96-well plate (4 × 10^4^ cells/well) and incubated for 24 h. Samples were dissolved in DMSO and pre-incubated with the cells at various doses for 1 h and then stimulated with LPS (1 µg/mL) for an additional 24 h. The cell culture supernatants were mixed with Griess reagent at a ratio of 1:1 and incubated at room temperature for 15 min. Absorbance at 540 nm was confirmed using a microplate reader.

#### 2.5.3. Enzyme-Linked Immunosorbent Assay

The cell culture supernatants were collected, and the levels of cytokines (TNF-α, IL-6, and IL-1β) and PGE_2_ were quantified according to the manufacturer’s instructions using an ELISA kit. The absorbance of the reacted samples was measured at 450 nm using a microplate reader.

### 2.6. Experimental Data Analysis

All of the data in this study were analyzed using SPSS software (version 20.0; IBM Corp., Armonk, NY, USA). Comparisons were performed using ANOVA with Duncan’s multiple range test and independence tests. A *p*-value < 0.05 was considered to indicate statistical significance.

## 3. Results and Discussion

### 3.1. Plant Materials Used in This Study

In this study, we isolated OFC (fresh callus before extraction and lyophilization) and OF (5 years old and dried cladode powder) (Figure 1).

### 3.2. Bioactive Compounds of OFC and of Extract

Cactus species are of scientific and commercial value and have a wide range of biologically active constituents. There have been reports of potential antioxidant activities in recently studied plant species. However, there is little information about saponin or flavonoid compounds. For this study, it was necessary to determine the biologically active compounds. Chromatograms of bioactive components identified in OFC and OF using UPLC-QTOF-MS are shown in Appendix A.

#### 3.2.1. Identification of OFC Metabolite Components

In OFC extract, 10 species (four triterpenoids, two fatty acids, one ester, and three steroids) were identified in the negative ion (NEG, Appendix A), and nine species (four triterpenoids, one antifungal, two fatty acids, and two fatty amides) were identified in the positive ion (POS, Appendix A). The metabolites identified in OFC consist of a chromatogram (Appendix A), a component index (Appendix A), and component fragments (Appendix A).

##### Triterpenoids

Peak 1 ([M-H]^−^) was identified as poricoic acid H (*m*/*z* 499.3430, C_31_H_48_O_5_). It has been reported that this compound activates apoptotic caspase-8 in human lung cancer A549 cells and transduces signals while cleaving the BH3 Bcl2-interacting protein [8]. Peak 1 ([M+H]^+^) was identified as polyporenic acid C (*m*/*z* 483.3470, C_31_H_46_O_4_). Peak 2 ([M-H]^−^) was identified as colossolactone VII (*m*/*z* 557.3480, C_33_H_50_O_7_). This compound is found in the fruiting bodies of Ganoderma colossum, mainly collected from the stems of *Delonix regia* (Fabaceae) in Thua Thien-Hue province, Vietnam, and it has been reported as an inhibitor of HIV-1 protease and SARS-CoV-2 major protease [9]. Peak 3 ([M-H]^−^) was identified as poricoic acid A (*m*/*z* 497.3273, C_31_H_46_O_5_). The combination of melatonin and poricoic acid A offers a therapeutic option for renal fibrosis during the AKI-CKD continuum [10]. Peak 4 ([M+H]^+^) was identified as corticatic acid D (*m*/*z* 481.3311, C_31_H_44_O_4_). Peak 5 ([M+H]^+^) was identified as corticatic acid B (*m*/*z* 465.3370, C_31_H_44_O_3_). Peak 6 ([M-H]^−^) was identified as poricoic acid C (*m*/*z* 481.3324, C_31_H_46_O_4_). Peak 7 ([M+H]^+^) was identified as 11,13,20,22-hentriacontatetraynoic acid (*m*/*z* 451.3569, C_31_H_46_O_2_).

##### Esters and Steroids

Peak 5 ([M-H]^−^) was identified as sootepin D (*m*/*z* 483.3481, C_31_H_48_O_4_). This compound has mainly been studied in the leaves of the Vietnamese plant *Gardenia philastrei* Pierre ex Pit, and anti-inflammatory effects have been reported [11]. Peak 6 ([M-H]^−^) was identified as cholesteryl hemisuccinate (*m*/*z* 485.3636, C_31_H_50_O_4_). Peak 9 ([M-H]^−^) was identified as pachymic acid (*m*/*z* 527.3733, C_33_H_52_O_5_). Triterpenoids such as pachymic acid are major components of *Poria cocos*, and anti-inflammatory, anti-cancer, and immunomodulatory actions have been reported [12,13]. Peak 10 ([M-H]^−^) was identified as eburicoic acid (*m*/*z* 469.3676, C_31_H_50_O_3_). These triterpenoid substances have the potential to protect the liver against CCl4-induced liver damage [14,15].

##### Antifungal

Peak 2 ([M+H]^+^) was identified as leptomycin B (*m*/*z* 541.3524, C_33_H_48_O_6_).

##### Fatty Acids

Peak 4 ([M-H]^−^)/Peak 4 ([M+H]^+^) were identified as LPE (18:2) (*m*/*z* 476.2784/478.2927, C_23_H_44_NO_7_P). Peak 5 ([M+H]^+^) was identified as LPC (18:2) (*m*/*z* 520.3399, C_26_H_50_NO_7_P). Peak 8 ([M-H]^−^) was identified as LPA (18:2) (*m*/*z* 433.2365, C_21_H_39_O_7_P). LPE (18:2) is very important for plant ripening and senescence [16]. It is known that the senescence-suppressing effect of LPE (18:2) inhibits the action of phospholipase D, which mediates the degradation of membrane phospholipids in the early stage of plant senescence [17]. The action of LPE (18:2) on phospholipase D increases according to the length and degree of unsaturation of the acyl chain contained in LPE (18:2). The aging-inhibiting effect of lysophospholipids (phospholipase D inhibition) is the greatest in LPE (18:2), while LPC (18:2) has a weak effect. LPA (18:2) promotes phospholipase D and shows the opposite tendency to LPE (18:2) or LPC (18:2) [18].

##### Fatty Amides

Peak 8 ([M+H]^+^) was identified as *cis*-11-eicosenamide (*m*/*z* 310.3101, C_20_H_39_NO). Peak 9 ([M+H]^+^) was identified as erucamide (*m*/*z* 338.3416, C_22_H_43_NO). Fatty acids are a source of metabolic and stored energy and are important to early growth development and metabolism [19].

#### 3.2.2. Network and Interpretation of Phytochemicals in OFC Extract

Saponins are secondary metabolites found in various plant species (such as beans, grains, seeds, bark, tea leaves, and root crops), and they protect against attack by potential pathogens [20,21]. Due to their toxicity to various organisms, saponins can be utilized for their antioxidant, antibiotic, fungicidal, anti-inflammatory, and pharmacological properties [22]. The biosynthetic pathway of saponins (a graphical abstract is provided), together with the major carbon source acetyl-CoA, produces a variety of derivatives essential for a number of cellular functions. Various secondary metabolites derived from the mevalonate (MVA) pathway affect sterol synthesis, growth, defense responses, and development [23]. On the other hand, MVA cannot exclude the interaction of the extrinsic pathway (2-C-methyl-D-erythritol 4-phosphate, MEP) with artificial culture. According to the study of *Arabidopsis thaliana* in medium, the rise in sucrose increased the substrate for MEP action by activity-induced SnRK1 (sucrose non-fermenting 1-related protein kinase 1) phosphorylation and the light-induced metabolism by light-induced IPP. It has been reported to increase MEP-derived terpenoids by increasing dimethylallyl pyprophosphate (DMAPP) production [24]. It can be said that biosynthetic patterns can vary depending on environmental conditions. In organisms, MVA is phosphorylated twice and then decarboxylated to form iso-pentenyl pyrophosphate (IPP), the basic backbone of isoprenoid biosynthesis [25]. Both IPP and DMAPP are used as synthetic intermediates. Then, squalene synthase, a key enzyme in the biosynthesis of the carbon ring backbone in the middle of triterpene saponins, catalyzes the internal cyclization of two molecules of farnesyl diphosphate to yield one molecule of squalene [26]. As triterpenoid glycosides, they have a 30C oxidosqualene precursor and are linked to glycosyl residues to form saponins [27,28]. Subsequent conversion into squalene forms various triterpenoid and fungal sterol compounds from 2,3-oxidosqualene. The physiological activity of the plant saponin is influenced by its chemical structure. Saponin glycosyl groups hydrolyzed by physical, chemical, and microbial transformations are converted into rare saponins (with short sugar chains) that are highly absorbed by the body. For example, it has been reported that saponin, an insoluble compound of ginseng, is converted by various biomaterials, and its physiological activity is modified [29]. However, it is clear that some saponins have side effects (allergies or gastrointestinal problems) when administered in high doses [30]. For cactus callus, it is believed that further research on component changes and mechanisms through additional processing is required.

#### 3.2.3. Identification of OF Metabolite Components

In OF extract, 11 species (two phenolic compounds, nine flavonols, one flavanone, and one flavone) were identified in NEG (Appendix A), and 12 species (one triterpenoid, nine flavonols, and two flavones) were identified in POS (Appendix A). The metabolites identified in OF are composed of a chromatogram (Appendix A), a component index (Appendix A), and component fragments (Appendix A).

##### Phenolic

Peak 1 ([M-H]^−^) was identified as 3-carboxy-4-hydroxy-phenoxy glucoside (*m*/*z* 315.0712, C_13_H_16_O_9_). Peak 2 ([M-H]^−^) was identified as piscidic acid (*m*/*z* 255.0502, C_11_H_12_O_7_). This compound is isolated from the rhizomes of *Cimicifuga racemosa* (Ranunculaceae).

##### Triterpenoids

Peak 3 ([M+H]^+^) was identified as astragaloside (*m*/*z* 641.1723, C_28_H_32_O_17_). This compound has been isolated and reported from *Astragalus membranaceus*, and its efficacy includes in colds, upper respiratory tract infections, fibromyalgia, and diabetes [31].

##### Flavonols

Peak 2 ([M+H]^+^) was identified as isorhamnetin 3-sophoroside-7-rhamnoside (*m*/*z* 787.2297, C_34_H_42_O_21_). This compound has been found in sea buckthorn (*Hippophaë rhamnoides* L.) [32]. Peak 3 ([M-H]^−^)/Peak 5 ([M+H]^+^) were identified as isoquercetin (*m*/*z* 463.0871/465.1036, C_21_H_20_O_12_). According to a recent study, it has the effect of improving brain damage in focal ischemia by linking anti-oxidant, anti-inflammatory, and anti-apoptotic effects [33]. Peak 4 ([M-H]^−^)/Peak 6 ([M+H]^+^) were identified as kaempferol 3-*O*-rutinoside (*m*/*z* 593.1497/595.1670, C_27_H_30_O_15_). Peak 4 ([M+H]^+^) was identified as rutin (*m*/*z* 611.1620, C_27_H_30_O_16_). Rutin (quercetin 3-*O*-rutinoside) is a bio-flavonoid widely distributed in vegetables and fruits. Due to its potent antioxidant and anti-inflammatory activity, it has shown excellent health benefits for extensive pharmacological applications in various anti-cancer studies [34]. Peak 5 ([M-H]^−^)/Peak 7 ([M+H]^+^) were identified as isorhamnetin-3-*O*-rutinoside (*m*/*z* 623.1598/625.1773, C_28_H_32_O_16_). Peak 6 ([M-H]^−^)/Peak 8 ([M+H]^+^) were identified as isorhamnetin 3-glucoside (*m*/*z* 477.1027/479.1189, C_22_H_22_O_12_). Peak 8 ([M-H]^−^)/Peak 9 ([M+H]^+^) were identified as quercetin (*m*/*z* 301.0343/303.0496, C_15_H_10_O_7_). Peak 9 ([M-H]^−^)/Peak 10 ([M+H]^+^) were identified as isorhamnetin (*m*/*z* 315.0500/317.0655, C_16_H_12_O_7_). This flavonol has a broad range of pharmacological effects, including cardiovascular protection; anti-inflammatory, anti-tumor, anti-oxidant, anti-bacterial, anti-viral, and anti-Alzheimer’s properties; and in neurodegenerative diseases [35]. Peak 10 ([M-H]^−^)/Peak 11 ([M+H]^+^) were identified as kaempferol (*m*/*z* 285.0390/287.0547, C_15_H_10_O_6_).

##### Flavanone

Peak 7 ([M-H]^−^) was identified as eriodictyol (*m*/*z* 287.0553, C_15_H_12_O_6_). This substance belongs to the flavanone family and is abundant in medicinal plants and citrus. The reported pharmacological effects include anti-oxidant, anti-inflammatory, anti-cancer, neuroprotective, cardioprotective, anti-diabetic, anti-obesity, and hepatoprotective effects [36].

##### Flavone

Peak 1 ([M+H]^+^) was identified as typhaneoside (*m*/*z* 771.2345, C_34_H_42_O_20_). Typhaneoside has been reported to prevent acute myelogenous leukemia by suppressing autophagy-related proliferation and inducing ferroptosis [37]. Peak 11 ([M-H]^−^)/Peak 12 ([M+H]^+^) were identified as hispidulin (*m*/*z* 299.0547/301.0705, C_16_H_12_O_6_). This aglycone has shown strong pharmacological effects, such as anti-oxidant, anti-fungal, anti-asthmatic, anti-inflammatory, and anti-convulsant effects, in several in-vitro studies, and it has been reported as a substance with anti-osteoporotic activity, proven to be similar to estrogen [38].

#### 3.2.4. Network and Interpretation of Phytochemical in OF Extract

The flavonoid compounds of each peak (including NEG and POS) were identified as glycones, except for eriodictyol, quercetin, isorhamnetin, kaempferol, and hispidulin, which are aglycones. The biosynthesis of secondary metabolites starts with the condensation of erythrose-4-phosphate and phosphoenolopyruvate to produce shikimic acid, a metabolite of aromatic amino acid biosynthesis [39]. Phenylalanine is converted into *trans*-cinnamic acid by PAL, and CA4H catalyzes the conversion to *p*-coumaric acid by the hydroxy group of the phenyl ring of cinnamic acid. The COOH of cinnamic acid forms a thioester bond with CoA, a process catalyzed by 4-coumarate-CoA ligase [40]. Chalcone synthase catalyzes the formation of chalcone by the condensation of one *p*-coumaric-CoA with three malonyl CoAs. Naringenin, a group of flavanones produced by CHI, is formed and converted into flavonols and flavones by the catalysis of F3H, FLS, and DFR, respectively. The phytochemical compounds of the cactus cladode have attracted attention as useful functional substances that can improve human health. Because of the high fiber content of cactus cladodes, trace compounds, such as phenolic and flavonoid compounds among phytochemicals, have not been studied in depth. On the other hand, in regions with tropical climates, including South America, it has been studied as a useful plant material for preventing metabolic diseases (obesity, diabetes, and cardiovascular disease). Traditional Mexican medicine recommends consuming cladodes due to the health effects of their bioactive compounds. The biological activities of cactus cladodes studied include anti-inflammatory, anti-diabetic, antibacterial, antioxidant, hyperlipidemic, neuroprotective, and immunoprotective activities [41]. The main antioxidant mechanism is hydrogen atom transfer by phenolic OH groups. The antioxidant activity of flavonoid compounds is influenced by specific chemical reaction patterns. The reaction patterns are the hydroxyl bond pattern and the catechol structure (substitution of 3′, 4′ OH groups of the B-ring), and they are also influenced by the level of conjugation sites allowed [42]. This flavonoid binding structure increases the degree of activity through resonance stabilization of radicals and induces interactions between compounds in the OF extract. The major flavonoid antioxidant activity has been reported at the levels of quercetin, kaempferol, and myricetin > rutin > luteolin > taxifolin by substitution and binding patterns. However, it is still unclear how to reflect absorption and metabolism with only known patterns [42,43]. To fractionate polymeric antioxidants from frozen and oven-dried cactus cladodes samples, acid hydrolysis treatment of the residue, sequentially extracted with methanol, acetone, and distilled water, increased antioxidant content by about eight times compared to conventional acid hydrolysis [44]. Additionally, the identification and detection of piscidic acid, quercetin, kaempferol, and isorhamnetin components were reported. They were similar to the phenolic and flavonoid compounds of cactus cladodes identified by LC-MS in this study.

### 3.3. Molecular Docking Results and Interpretation

To predict the overall physiological activity of each extract, some specific anti-inflammatory compounds were selected from the metabolites profiled. The selected compounds were subjected to molecular docking assays to confirm their binding interactions with inflammatory response receptors. Toll-like receptor 4 (TLR4) and mitogen-activated protein kinase (MAPK) are important receptors in the LPS-induced inflammatory response. In general, cell signaling pathways in the regulation of inflammation are complex and involve cross-talk, but the interactions between phytochemicals and protein receptors are related to their bioactivities. The interaction results between the target protein and the bioactive compound ligand selected from OFC and OF extracts are as follows (Appendix A). Docked saponin compound ligands showed binding energies ranging from −6.2 to −6.6 Kcal/mol for the TLR4 protein and binding energies ranging from −7.4 to −8.3 Kcal/mol for the MAPK protein. High affinity was shown for colossolactone VII (−6.6 Kcal/mol) in the TLR4 protein and poricoic acid H (−8.3 Kcal/mol) in the MAPK protein. Docked flavonoid compound ligands showed binding energies ranging from −6.5 to −7.6 Kcal/mol on the TLR4 protein and binding energies ranging from −7.3 to −9.0 Kcal/mol on the MAPK protein. High affinity was shown for rutin (−7.6 Kcal/mol) in the TLR4 protein and rutin (−9.0 Kcal/mol) in the MAPK protein. These results indicate that the affinity with MAPK protein is greater than the affinity with TLR4 protein. A higher negative value (affinity or binding energy) in a docking study indicates greater affinity between the target protein and the ligand molecule, indicating greater efficiency of the bioactive compound. The affinity between a protein and a ligand for the van der Waals term is known to be about −15.0 Kcal/mol at most in biophysical binding, and about −9.0 Kcal/mol has been interpreted as a normal distribution [45]. The visualized molecular docking 3D interaction diagram is shown in Figure 2 and Figure 3. Complex compounds extracted from plants are considered predictable only when many variables and diverse of compound types are systematically calculated to confirm their applicability to the human body before being used as food through molecular dynamics or docking analysis. To understand the plausible experimental anti-inflammatory activity of the compounds currently studied, prediction of inhibitory dissociation constant (Ki) values was performed. The predicted Ki (pKi) value indicates how potent a compound is as an inhibitor of a biological process, and it is used as a universal number to symbolize the concentration required to produce half-maximal inhibition [46,47]. The pKi values of the selected saponin compounds for OFC extracts were in the range of 14.53–28.53 µM for TLR4 and 0.82–3.77 µM for MAPK. Selected flavonoid compounds for the OF extract showed TLR4 in the range of 2.69–17.20 µM and MAPK in the range of 0.25–4.46 µM. Each compound in the extract was correlated with the binding energy. The pKi values obtained clearly demonstrated the plausible high inhibitory potential of the presently investigated compounds with MAPK domains.

### 3.4. Anti-Inflammatory Activity of OFC and of Extract

In-vitro anti-inflammatory activities of OFC and OF extracts were compared. RAW 264.7 cells were treated with two samples (OFC and OF in 70% ethanol extracts) at various concentrations for 24 h, and their cytotoxicity was evaluated using the EZ-Cytox reagent. No cytotoxicity was observed in either sample up to a concentration of 200 μg/mL (Figure 4A). The concentrations of the two samples used in the activity experiment were determined to be 25, 50, 100, and 200 μg/mL. LPS treatment showed a significant decrease in NO production in a dose-dependent manner (*p* < 0.001, Figure 4B). In addition, in the comparison between the two samples, the activity of OF was significantly lower than that of OFC (*p* < 0.001). The production amount of PGE_2_ was significantly increased compared to the control group (*p* < 0.001). In the comparison of the samples, OFC was 9.35 ng/mL, OF was 8.44 ng/mL at a concentration of 200 μg/mL, and the production of OFC was significantly decreased (*p* < 0.01). In addition, LPS-treated cells produced pro-inflammatory cytokines, such as TNF-α, IL-6, and IL-1β, compared to the control group. On the other hand, NO, PGE_2_, and pro-inflammatory cytokines (TNF-α, IL-6, and IL-1β) increased by LPS were significantly decreased in a dose-dependent manner in cells pretreated with each sample extract (*p* < 0.05, Figure 4C–F). Pro-inflammatory cytokine factors at a concentration of 200 μg/mL included LPS-stimulated TNF-α (OFC 72.33 ng/mL, OF 66.78 ng/mL) and IL-1β (OFC 49.10 pg/mL, OF 34.45 pg/mL), both of which significantly decreased OF (*p* < 0.01, *p* < 0.001). In particular, at a concentration of 50 μg/mL or more, cacti reduced the production of IL-1β by LPS stimulation more effectively than OFC (western blotting results for expression changes are not shown). In addition, the anti-inflammatory effects of cacti have been reported in the past [48]. 

## 4. Conclusions

This study showed that OFC and OF extracts grown in Korea can be considered secondary metabolite sources involved in the inhibition of LPS-induced inflammatory responses. A qualitative analysis allowed for the identification of saponin and phenolic compounds related to triterpenoids and flavonol glycosides. Exploration of the biological activity of the studied extracts showed that OFC and OF possess antioxidant and anti-inflammatory effects. From the results obtained, it seems feasible to use OFC and OF as sources of natural compounds that could be included in foods, cosmetics, and pharmaceuticals. These findings could stimulate research to elucidate the mechanism of action of plant saponins and flavonoids with similar structures, including those with antioxidant and anti-inflammatory properties.

## Figures and Tables

**Figure 1 antioxidants-12-01329-f001:**
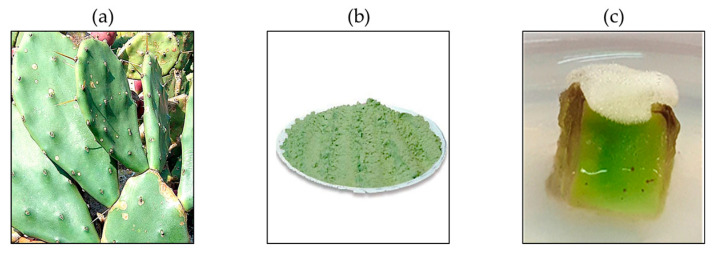
The characteristics of OFC and OF used in this study: (**a**) fresh cactus cladodes (*Opuntia ficus indica*); (**b**) dried cactus cladode powder; (**c**) fresh callus.

**Figure 2 antioxidants-12-01329-f002:**
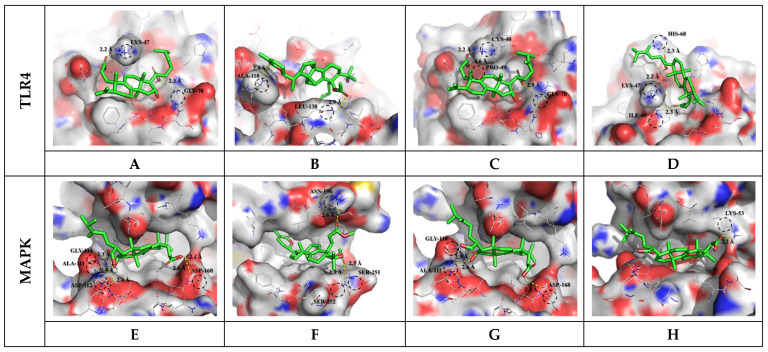
Molecular docking study of selected saponin compounds (Compound CIDs: 10918099, 24898464, 5471851, and 56668247) and of human TLR4 (PDB ID: 2Z62) and MAPK (PDB ID: 5MTY). Molecular docking of poricoic acid H (**A**,**E**), colossolactone VII (**B**,**F**), poricoic acid A (**C**,**G**), poricoic acid C (**D**,**H**), and LPS to TLR4 and MAPK receptor.

**Figure 3 antioxidants-12-01329-f003:**
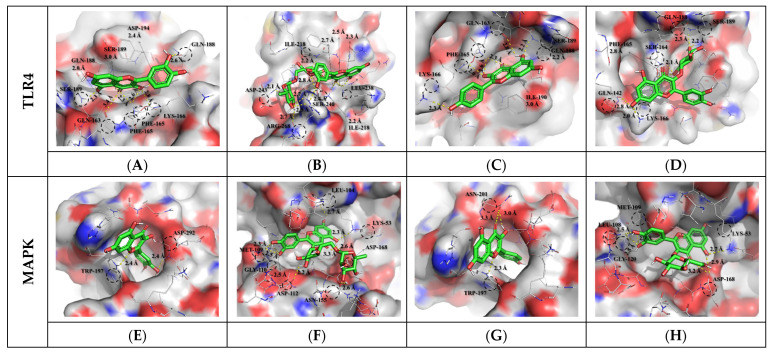
Molecular docking study of selected flavonoid compounds (Compound CIDs: 5280343, 5280805, 5280863, and 5280804) and of human TLR4 (PDB ID: 2Z62) and MAPK (PDB ID: 5MTY). Molecular docking of quercetin (**A**,**E**), rutin (**B**,**F**), kaempferol (**C**,**G**), isoquercetin (**D**,**H**), and LPS to TLR4 and MAPK receptor.

**Figure 4 antioxidants-12-01329-f004:**
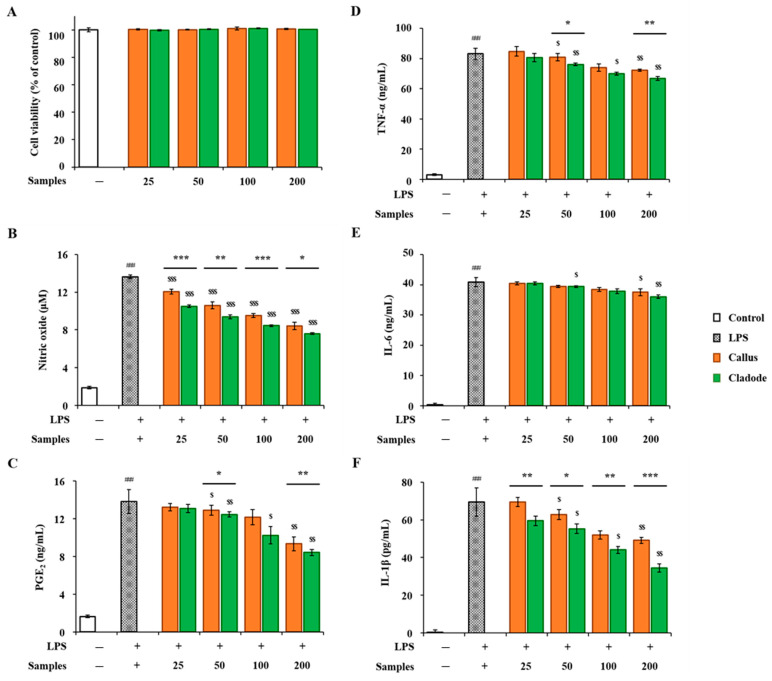
Anti-inflammatory activities of OFC and OF extract on LPS-stimulated RAW 264.7 macrophages: (**A**) cell viability; (**B**) nitrate oxide; (**C**) PGE_2_; (**D**) TNF-α; (**E**) IL-6; and (**F**) IL-1β. The data are shown as the mean ± SD (n = 3). ### *p* < 0.001 compared with the control (none) group; * *p* < 0.05, ** *p* < 0.01, and *** *p* < 0.001 compared with LPS-stimulated RAW 264.7 cells and treatment groups; $ *p* < 0.05, $$ *p* < 0.01, and $$$ *p* < 0.001 compared by sample group.

## Data Availability

Data are contained within the article and Appendix A.

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
