# Peer review of "The Cactus (Opuntia ficus-indica) Cladodes and Callus Extracts: A Study Combined with LC-MS Metabolic Profiling, In-Silico, and In-Vitro Analyses"

_antioxidants, 2023, doi:10.3390/antiox12071329_

Round 1
Reviewer 1 Report
I feel that one of the most significant findings in your study is the observation that callus tissues may overexpress the mevalonate secondary metabolism pathway vs the shikimate pathway in cladodes, as documented by the different metabolites detected. Therefore, ths is worth mentioning in more detail under 3.2. along with some suggestions for future research to elucidate the origin of the different metabolic pathways.
Author Response
First of all, we would like to thank you and all the reviewers for going through our manuscript and making the needful comments with respect to the manuscript.
We made all the changes as suggested by reviewers and tried our best to give appropriate answers for the queries raised by them.
Further, we would like to bring it to your kind attention that the changes made in the manuscript are indicated with color or memo.
If additional anything is needed to be addressed, kindly let us know.
I eagerly await your response.
Sincerely,
(Dong-Geon, Ph.D.)
First and Corresponding author
* Please see the attachment.

Reviewer 2 Report
Dear authors,
After the review process, I have several comments: the abstract should contain numerical data; the last phrase from the introduction should be deleted; it is for the conclusion maybe - Line 59; Figure 1 should be a supplementary file; Table 1, 2 & 3 - supplementary file; Figure 2 should be deleted or used as graphical abstract; figure 5 should be moved until the section 3.4; the study should have negative points and future application of the study.
Best regards!Minor modifications should be realized, nothing important.
Dear authors,
After the review process, I have several comments: the abstract should contain numerical data; the last phrase from the introduction should be deleted; it is for the conclusion maybe - Line 59; Figure 1 should be a supplementary file; Table 1, 2 & 3 - supplementary file; Figure 2 should be deleted or used as graphical abstract; figure 5 should be moved until the section 3.4; the study should have negative points and future application of the study.
Best regards!
Author Response

(The authors gave the same response as above.)

Reviewer 3 Report
The manuscript entitled “The cactus (Opuntia ficus-indica) cladodes and callus extracts: A study combined with LC-MS metabolic profiling, in-silico, and in-vitro analysis” investigated the phytochemical profiling and biological activity of cactus and callus extracts.
The manuscript is quite well-written and the results are considerable. However, to be published by Antioxidants, authors should seriously consider the following points:
1. The present study is submitted to the Antioxidants journal, however, the manuscript lacks information on antioxidants. This keyword should be used more throughout the manuscript, especially in the abstract and conclusion.
2. Abstract and Conclusion must be rewritten. The statements are too general. The authors should highlight the main findings and novelty of this study.
3. Those identified compounds reported with antioxidant capacity (in previous studies), should be mentioned in the discussion. In addition, a reasonable explanation of the correlation between functional groups in the structure of phenolics and antioxidant activity should be included in the discussion. This document is recommended for reference and citation 10.3390/molecules24030605.
4. Materials and methods:
· A figure of plant samples is required
· Specimen voucher number, identification of OF, and OFC’s information (e.g. age? How many days? week?) should be provided
· Reservation condition of samples after extraction should be mentioned (if applicable)
· UPLC: mobile phase program (linear or gradient?) must be supplied
5. Results and Discussion:
· L157: “…The sclera of Poria cocos Wolf (Polyporaceae) is traditionally used..”. It does not make sense when this sentence appears here.
· L184-187: Is there a reasonable explanation for the classification of this compound, antifungal?
· Figure 2 is not a result of this study. It can be summarized in the text with appropriate citations instead of being a figure.
· Discussion should go in accordance with the results. Authors should properly choose a descriptive or inductive writing style.
6. References should be shortened. Some old papers can be deleted.
To sum up, the authors should extensively revise the manuscript focused on highlighting the novelty and originality of the present study. I suggest a major revision for this manuscript.
English writing should be extensively improved. An English editing service with corrections from native speakers is highly recommended.
Author Response

(The authors gave the same response as above.)

Round 2
Reviewer 2 Report
No other comments.
Author Response
Thank you for your guidance.
Please see the attachment.

Reviewer 3 Report
The authors addressed most of my comments, however, I'd like to suggest some more points as follows:
1. Introduction of Opuntia ficus-indica (rationale) should be added.
2. The authors have not supplied the information of UPLC mobile phase program. If it was linearity, What was the ratio between A and B solvents?
3. I totally agree with the authors about the synergic contribution of compound combinations to the antioxidant activity of extracts. However, I still suggest adding the contributory role of functional groups because this has been validated in many chemical and pharmaceutical studies.
Also, if you wanna use your personal thinking in the academic discussion, you must show strong evidence (proper citations or references). In this case, the citation with ref. 43 and 44 did not match with your personal thinking about answering "Point 3" in my previous comments.
4. Conclusions must be rewritten focusing on new findings, significance and future aspects. Reference should not appear in the conclusion section.
I recommend one more round for major revision.
Moderate editing of English language required
Author Response

(The authors gave the same response as above.)

Round 3
Reviewer 3 Report
The authors almost addressed all the recommended issues.
A minor revision is suggested.
L158-168: About "Introduction of Opuntia ficus-indica": properly revise and then move to the introduction part.
Minor editing of English language required
Author Response
Thank you for reading my thesis and for detailed guidance.
It's been a good opportunity.
Thank you so much.
Please see the attachment.
